# A Drug Discovery Approach to a Reveal Novel Antioxidant Natural Source: The Case of Chestnut Burr Biomass

**DOI:** 10.3390/ijms25052517

**Published:** 2024-02-21

**Authors:** Alfonso Trezza, Michela Geminiani, Giuseppe Cutrera, Elena Dreassi, Luisa Frusciante, Stefania Lamponi, Ottavia Spiga, Annalisa Santucci

**Affiliations:** 1Department of Biotechnology Chemistry & Pharmacy, University of Siena, Via A. Moro, 53100 Siena, Italy; geminiani2@unisi.it (M.G.); giuseppe.cutrera@student.unisi.it (G.C.); elena.dreassi@unisi.it (E.D.); luisa.frusciante@unisi.it (L.F.); stefania.lamponi@unisi.it (S.L.); ottavia.spiga@unisi.it (O.S.); annalisa.santucci@unisi.it (A.S.); 2SienabioACTIVE, University of Siena, Via A. Moro, 53100 Siena, Italy

**Keywords:** drug and target discovery, molecular modeling, docking and molecular dynamics simulation, chestnut burr, natural bioproduct, bioeconomy, polyphenols, antioxidant

## Abstract

Currently, many environmental and energy-related problems are threatening the future of our planet. In October 2022, the Worldmeter recorded the world population as 7.9 billion people, estimating that there will be an increase of 2 billion by 2057. The rapid growth of the population and the continuous increase in needs are causing worrying conditions, such as pollution, climate change, global warming, waste disposal, and natural resource reduction. Looking for novel and innovative methods to overcome these global troubles is a must for our common welfare. The circular bioeconomy represents a promising strategy to alleviate the current conditions using biomass-like natural wastes to replace commercial products that have a negative effect on our ecological footprint. Applying the circular bioeconomy concept, we propose an integrated in silico and in vitro approach to identify antioxidant bioactive compounds extracted from chestnut burrs (an agroforest waste) and their potential biological targets. Our study provides a novel and robust strategy developed within the circular bioeconomy concept aimed at target and drug discovery for a wide range of diseases. Our study could open new frontiers in the circular bioeconomy related to target and drug discovery, offering new ideas for sustainable scientific research aimed at identifying novel therapeutical strategies.

## 1. Introduction

In the recent five decades, the world population increased from 2.5 billion to 7.9 billion people [1]. This spread could easily be explained by the relationship between human needs and the environment’s availability and abundance. The basis of this relationship is represented by linear economics, where using and exhausting natural resources and primary energy, with a non-substitutable bioeconomic concept, allowed for a rapid spread of the world’s population [1]. Unfortunately, uncontrollable and irresponsible consumption has represented a critical element in the serious social, economic, and environmental crisis we are experiencing globally, causing a near depletion of natural resources.

All capital forms are related to the natural capital, which is not substitutable, by definition; however, Yardley and Dewulf et al. support the idea that “we received this world as an inheritance from past generations, but also as a loan from future generations” [2]. Thus, we must attempt to minimize the impact of human activities on the natural environment, human health, and natural resources [2]. Hence, this encompasses the idea of switching from a linear economy to a circular economy to solve and heal the planetary emergency.

The circular economy is aimed at decreasing the consumption and then the depletion of natural sources by increasing the reuse and the recycling of the resource [3]. The circular economy is developed within a wide range of applications, such as the bioeconomy, which is a circular economy applied to scientific research fields—leading to the definition of circular bioeconomy [4].

The circular bioeconomy is based on an economic model that utilizes renewable and reusable natural resources to drastically reduce waste, which may also represent a pollution source [5]. In contrast to the linear economy, the circular bioeconomy aims both at using (for as long as possible) and recovering a natural source, offering an approach to convert and take advantage of our agricultural and industrial systems to obtain sustainable products, thus avoiding altering the natural balance of our planet.

In recent years, the circular bioeconomy has successfully been developed and applied in different fields, and important scientific and technological milestones have been achieved [6]. Interestingly, the circular bioeconomy application on biological waste is increasingly used, attracting considerable interest. Such application is aimed at using different natural waste kinds, such as food waste, industrial waste (i.e., paper, wine, and beer industries), and agriculture/forestry waste to obtain bioproducts for different endeavors [7].

Despite the huge progress made in this field, the application of the circular bioeconomy is sometimes complicated, representing a challenge for the scientific community.

*Castanea sativa* (*C. sativa*), or sweet chestnut (as known in Europe), is a tree with a high commercial and economic impact in south Europe due to its ability to produce valuable fruits. The *C. sativa* tree represents a precious resource in mountainous regions, providing “nuts” characteristic (chestnuts) of the Mediterranean diet with important organoleptic and pro-healthy properties [8,9].

Chestnuts are the edible seeds of trees and shrubs belonging to the genus Castanea; however, the most widely known and cultivated chestnut is the sweet chestnut (*C. sativa*). Chestnuts have a shiny, dark brown outer shell or husk that is prickly and split into several sections. Inside the husk, there is a smooth, glossy brown nut with a distinct pointed tip. The size of chestnuts can vary, but they are generally about the size of a golf ball. The outer shell is hard and spiky, providing protection for the nuts inside. The nuts themselves have a smooth, shiny surface. The flesh of chestnuts is starchy and has a sweet and nutty flavor when cooked. Chestnuts have a unique flavor that is slightly sweet and nutty. They are often compared to other nuts but are less oily and have a distinct taste.

Different parts of C. sativa have long been used in traditional medicine for different disorders, like cough, diarrhea, and infertility [10].

In fact, chestnuts are excellent dietary sources that provide carbohydrates, fiber, starch, fatty acids, minerals (mainly potassium, phosphorus, calcium, and magnesium), and vitamins (B9, C and E) [8,11]. In Europe, the chestnut sector has shown growing interest over recent years, leading to high investments being made from governments, industries, and factories to this sector [8]. Spain, Italy, France, Brazil, United States of America (USA), and Switzerland represent the countries with the highest production of chestnuts in exportation terms (77.6%), suggesting the scale of importance of these industries’ economic revenue [12].

*C. sativa* represents one of the most economically important fruit crops in southern Europe, particularly on the northern side of the Mediterranean basin, but only its bark and wood chips are used to extract tannins. The bark of *C. sativa* contains approximately 60% of active tannic substances, composed mainly of castalagin, vescalagin, castalin, and vescalin, which are all easily hydrolysable glycosidic tannins [8].

Although there are indications that chestnuts are a rich source of tannins [9,10], most studies have evaluated the use of leaves, bark, and wood [8,10,11], and scarce information is available about the potential use of other types of chestnut waste, e.g., chestnut burrs. Studies on chestnut by-products revealed a good profile of bioactive compounds with antioxidant, anticarcinogenic, and cardioprotective properties [13].

In Europe, chestnuts are considered a Mediterranean crop and are facing significant challenges due to the rising temperatures and increased drought conditions associated with climate change. These factors have a profound impact on the wellbeing of the species, making it more susceptible to severe attacks by *Dryocosmus kuriphilus*. The necessary removal of chestnut burrs from brushwood land is crucial for ensuring optimal chestnut tree growth. However, this task often requires a significant investment of time and financial resources, leading to a common and environmentally detrimental practice of burning the burrs, contributing to the emission of carbon dioxide.

In this work, we first characterized a hydroalcoholic extract obtained from *C. sativa* chestnut burrs and then applied a computational workflow by providing innovative and integrated silico and in vitro approaches to identify antioxidant natural compounds and their potential biological targets.

Then, we obtained the entire receptor complex involved directly or indirectly in the cell oxidative stress onset, a pathological condition present in several disorders [14], including rare diseases [15,16]; finally, we identified the best potential targets of identified compounds (gallic acid, quinic acid, protocatechuic acid, brevifolin carboxylic acid, and ellagic acid). Classical and steered molecular dynamic simulations elucidated pharmacodynamics/kinetics features into the target/compound interaction to further validate our study.

This approach could represent a starting point for a new application of the circular bioeconomy to identify natural compounds from natural waste, targeting biological systems involved in the onset of pathologies; significantly reducing time, cost, and chemical waste; and preserving the failing impact on the environment.

## 2. Results

### 2.1. LC-DAD-MS Analyses

The chromatographic profile can be seen at 254 nm, and the trends obtained by extracting the contribution of the pseudo-molecular ions of the identified chemical species (Appendix A) (signal in negative [M − H]^−1^) for the two samples analyzed are displayed in Figure 1. The LC-DAD-MS analysis of the extract allowed us to identify five different compounds: gallic acid, quinic acid, protocatechuic acid, brevifolin carboxylic acid, and ellagic acid (Appendix A). Most of the spectral information was obtained in negative mode due to the acidic characteristics of the identified molecules. Amongst the identified compounds, gallic acid and ellagic acid are the most abundant components in the extract.

### 2.2. In Silico Results

#### 2.2.1. Target/Compound Virtual Screening

To define the potential targets involved in the interaction with our compounds, a ligand-based virtual screening was performed against the whole oxidative stress target complex identified through the “target section” in the DrugBank database [17]. Our research provided 212 targets involved indirectly/directly in the cell oxidative stress condition, and each primary structure was downloaded from the UniProt database. To verify the availability of target 3D structures, a multiple sequence alignment (MSA) was performed through BLASTp with the “Protein Data Bank” database [18]. From the MSA results, we obtained and downloaded a total of 103 protein 3D structures (Appendix A).

Each structure was optimized though molecular modeling, solving potential structural gaps and steric clashes; then, a virtual screening was performed among all targets and gallic acid, quinic acid, protocatechuic acid, brevifolin carboxylic acid, and ellagic acid. To standardize our analyses, and to add strength and reliability to our in silico results, we adopted two different strategies to select the best five complexes—(i) binding free energy (docking score) and (ii) evolution approach (considering the interaction network consensus binding residues), as suggested in a previous work [19]. We selected the first five complexes with the highest binding free energy: Death-associated protein kinase 1 (k1), cyclin-dependent kinase 2 (k2), MAP kinase-activated protein kinase 3 (k3), Cyclin-dependent kinase 5 (k4), and mitogen-activated protein kinase 10 (k5). Remarkably, all targets were in the same complex with ellagic acid, showing a binding free energy from −9.5 Kcal/mol to −8.9 kcal/mol. Interaction network analyses showed that the ellagic acid formed a wide polar and hydrophobic interaction network within the target binding pocket (Figure 2). Furthermore, MSA results revealed the ability of our compound to trigger strong polar interactions with target key residues: Ile-10 (ATP binding site), Lys-33 (ATP binding site), and Asp-144 (DFG motif) (residue number based on human cyclin-dependent kinase 5 (k4) with UniProtKB entry—Q00535-) (Appendix A). Taken together, our results suggest the ability of our compound to potentially inhibit the targets upon the occurrence of strong interactions with critical residues. Based on such evidence, the best five complexes were selected for further in silico analyses to dissect and define the pharmacodynamic/kinetic features of target/compound complexes.

#### 2.2.2. cMD: Target/Ellagic Acid’s Docking Pose Stability and Interaction Energy

To further explore the target/compound interaction and to define structural and energy features, a cMD was performed for each complex. To avoid potential computational bias, we evaluated the structural integrity of the target backbone for each complex. RMSD analyses suggested a good structural stability, showing a RMSD stable trend between 0.2 Å and 0.7 Å (Figure 3). The binding poses of the compound within the target binding pocket did not significantly show fluctuations during the MD run, providing a RMSD range between 0.05 Å and 0.1 Å (Figure 3), suggesting good stability and reliability of the selected starting docking pose. To further strengthen the impact of the docking and cMD simulation, target/compound interaction energy was evaluated. From energy analyses, we noticed that our compound was able to spontaneously bind on the target with a very high total interaction energy—between −69.5 ± 1.3 kcal/mol and −47.5 ± 3.7 kcal/mol (Appendix A).

#### 2.2.3. SMD: Target/Ellagic Acid’s Unbinding Pathway

Simulations were carried out using steered molecular dynamics (SMD) to analyze the unbinding path of our compound from the target. The compound reported a very different steady increase of the applied forces on the first ~150 and ~650 ps of the simulation for each target, until they reached the maximum, called rupture force; then, the force quickly decreased to 0, showing the total unbinding of ligand from the target. In brief, the force in the first step was between 0 and 150 ps for k1, 0 and 350 ps for k2, 0 and 370 ps for k3, 0 and 650 ps for k4, and 0 and 350 ps for k5, which made the compound slowly detach and move away from the binding region; in the second step, between the rupture force time of 320 ps (k1), 350 ps (k2), 380 ps (k3), 660 ps (C4), and 240 ps (k5) of the simulation, the compound moved away from the target and entered the solvent region (Figure 4).

## 3. Discussion

The development of the bioeconomy and the implementation of green forest management play a key role in combating climate change and enhancing the value of the nation’s natural capital. The latest trends in national agricultural indicators reveal an upswing in the overall economic value of the primary sector, particularly driven by biological and ‘Protected Geographical Indication’ productions. Chestnut (*Castanea sativa* Miller) cultivation has a longstanding tradition in Italy, serving as the main staple food for the population in hilly mountainous areas for centuries. About 52,356 tons of it were produced by Italy, the main chestnut producer in the European Union, which provided 38% of the total European chestnut production in 2017. (FAOSTAT, food and agriculture organization of the United States: http://www.fao.org/faostat/en/#data, accessed on 17 July 2019). Nevertheless, over the recent few decades, chestnut production has declined due to diseases, such as chestnut blight and chestnut gall wasp, along with the progressive depopulation of mountainous areas. Given the difficulties that the chestnut sector is facing, a new approach to generate value-added products, not only from the fruit but also by exploiting other components of the plant, emerges as an important strategy to promote value chain development and attract investments in an efficient and effective manner. *C. sativa* serves as a common source of polyphenols, primarily tannins. These tannins are in the form of hydrolyzable tannins, including ellagitannins and gallotannins, which are present in the bark, flours, and leaves. In addition, condensed tannins, specifically procyanidins, are exclusively found in chestnut peels. Notably, the main ellagitannins identified in the bark are vescalagin and castalagin [12,20]. The chestnut peeling processes generate solid waste, consisting of burrs, which accounts for about 10–15% of the whole chestnut weight. Currently, chestnut burrs are primarily used as fuel or exploited as a source of fermentable sugars for biofuel production [19]. Additionally, the practice of burning chestnut burrs poses an environmental problem, as it may generate several toxic compounds like dioxin [21] and generate CO_2_, whose increase is considered one of the main causes of climate change. The results obtained in this work revealed the high potential of this chestnut by-product as a new source of bioactive compounds, which might provide a novel strategy to stimulate the application of waste products as a new supplier of this type of compounds.

We developed a comprehensive computational strategy to identify potential targets involved in oxidative stress conditions and bioactive compounds extracted via chestnut burr biomass. Firstly, the entire oxidative stress target complex was obtained through the “target section” implemented in the DrugBank database; then, molecular modeling was performed to obtain and optimize the 3D structures of the targets that were previously selected. To define the potential compound(s) extracted via chestnut burr biomass with antioxidant proprieties and its biological target(s), a virtual screening of bioactive compounds was performed, with each target being tested through computational virtual screening. As a result of the docking simulation, ellagic acid was determined as the compound with the highest binding free energy score of all compounds on the best five targets, triggering strong polar interactions with target critical residues [22,23]. Remarkably, the experimental characterization with LC-MS analyses showed ellagic acid being the most abundant component in the extract. The docking simulation is a valid method to define the ability of a compound to bind against a target; however, it suffers from major limitations, which are mostly related to the static or semi-flexible treatment of ligands and targets. To overcome the limitations of the docking simulation and provide robustness to our analyses, we combined cMD and SMD simulations to explore and dissect further molecular insights. cMD and SMD simulations confirmed the ability of ellagic acid to strongly bind on targets by identified us, exhibiting a binding pose that is very stable within the binding pocket. Interaction energy analyses performed between ellagic acid and its targets further confirmed our analyses, showing significant interaction energy values with a very low error estimation along the entire MD run.

Interestingly, in SMDs, we observed an initial slow detachment phase and a second phase where the compound moved away from the target. The biological significance of these variations in unbinding times and forces among different targets could be explained due to the different starting interaction network of the ligand within the target binding pocket. The compound shared a similar binding pose and pocket; however, the starting binding pose (Figure 2) showed differences in the interactions triggered between the compound and the binding residues. Such differences would lead the ellagic acid to show different pull force values against the targets. This variability could affect the compound’s potential efficacy on targets because a higher pull force value would lead to a slower dissociation of the ligand from the target, enhancing the compound’s potential efficacy.

Furthermore, the interaction energy analyses and the time-averaged force profiles during the unbinding simulations of computed results for each complex provided both energy values and unbinding forces that were very high, confirming the reliability of the docked pose selected by us and in silico studies.

Altogether, the simulations strongly reinforced the evidence provided in our study, and reliable structural and energy information about the ability of compounds to interfere with the target biological function was shown.

Remarkably, previous in silico and in vitro works that reported on the antioxidant activities of ellagic acid targeting the kinase family showed that they share the same binding pocket and a similar interaction network and energies with binding residues to the ones proposed by us [24,25,26]. Furthermore, Yusuke et al. determined the 2.35Å crystal structure (PDB code 2ZJW) of a human CK2 catalytic subunit in a complex with ellagic acid [27]. Remarkably, both the binding pocket and the binding mode of the ellagic acid in 2ZJW was extremely similar to our docking results (Appendix A), confirming the validity and reliability of our bioinformatics approach.

## 4. Materials and Methods

### 4.1. Materials and Samples

Burrs from *C. sativa Mill*, certified as Castagna del Monte Amiata IGP (Reg. Cee n. 2081/92), were obtained from the “Associazione per la Valorizzazione della Castagna del Monte Amiata IGP” in Tuscany, and then they were collected. After being dried at room temperature, the samples were pulverized in a laboratory homogenizer before being stored in a dark, sealed plastic bag at −80 °C in a dark environment until extraction.

All chemicals were obtained from Sigma-Aldrich (St. Louis, MO, USA).

### 4.2. C. sativa Chestnut Burrs Extract Preparation

In total, 100 mL of ethanol/water 70/30 (*v*/*v*) was mixed with 10 g of chestnut burr powder, resulting in a solid-to-solid ratio of 1/10 *w*/*v*. A thermostatic mantle was employed to stir the mixture for 3 h at 80 °C. The extraction chamber was connected at the top to a net water-cooled bubble condenser, allowing the evaporated solvent to reflux. Subsequently, the mixture was centrifuged (30 min, 4000 rpm), and the supernatant was separated from the residual biomass and filtered. The organic solvent was removed using a rotavapor, while the aqueous residue was freeze-dried to obtain the dry extract.

### 4.3. LC-DAD Analysis

LC analysis of the chemical composition of chestnut burrs extract was carried out on an Agilent 1260 LC/UV-DAD (Agilent Technologies, Palo Alto, CA, USA) equipped with a binary pump delivery system, a degasser, and UV-DAD detector. Based on the chemical properties of the analytes, UV monitoring was performed at 210 and 254 nm. A C18 Luna column (250 mm × 3.0 mm, 5 µm particle size) (Phenomenex, Torrance, CA, USA) was used. A mobile phase composed of water/formic acid (99.5/0.5 *v*/*v*) (solvent A) and acetonitrile (solvent B) was used.

For MS detection, N2 was used as a nebulizing and drying gas; the vaporization temperature as well as the capillary and fragmentor voltage were set at 350 °C, 3000 V, and 70 V, respectively. Mass spectra were acquired over the scan range of 100–1500 *m*/*z* in both positive and negative mode.

Extract was obtained as reported above, was solubilized (1 mg/mL of acidified mobile phase (A/B 1/1 *v*/*v*), and was injected for the identification of principal components.

The following gradient elution was applied—from 0 to 2 min, 0% B; from 2 to 10 min, 20% B; and from 10 to 20, 85% B, and remained so for 21 min. The flow rate was 0.5mL/min. The injection volumes were set to 20 μL. The identification of the principal components was carried out based on UV profiles, mass spectra, and a comparison with retention time, which comprised standard solutions of gallic, quinic, protocatecuic, brevifolin carboxylic, and ellagic acid.

### 4.4. Structural Optimization and Resources

The oxidative stress target complex was retrieved in DrugBank [16] using “target section” with the key word “oxidative stress”. Their 3D structures and FASTA sequences were retrieved from the RCSB Protein Data Bank [28] and UniProt database [29], respectively. The 3D structures were obtained by performing a multiple sequence alignment with BLASTp v.2.15.0 and choosing PDB as the search database; all parameters were used as default [17]. To further confer robustness and reliability to our docking studies, we selected only the targets with 3D structures that were combined with their inhibitor. This resulted in 103 complexes being available on the Protein Data Bank (Appendix A).

To avoid errors during molecular dynamic (MD) simulations, the potential missing side chains and steric clashes in 3D structures reported in PDB files were added/resolved with molecular/homology modelling using MODELLER v.9.3 implemented in PyMOD3.0 (PyMOL2.5 plugin) [30]. Three-dimensional structures were analyzed and validated with PROCHECK v.3.5.4 [31]. GROMACS 2019.3 [32] with charmm36 force field [33] was used to minimize the high-energy intramolecular interaction before the docking simulations were carried out, and CHARMM-GUI v.3.8 [34] was used to assign all parameters to the biological targets and ligands.

In detail, prior to the performance of further simulations, the starting conformation sequence was aligned against its primary structure; thus, the potential missing side chains were added to the structure. Furthermore, loop modeling implemented in MODELLER v.9.3 was used to optimize the best starting orientation of each loop of the structure. Lastly, each structure was analyzed in the PROCHECK tool, where a Ramachandran plot (which analyzes the backbone of ϕ and ψ angles and Chi1–Chi2 plots for side chains) confirmed the validity of the starting conformation. Then, we minimized the energy of each structure by performing energy minimization using GROMACS 2019.3 with the charmm36 force field. We did this to prevent the possibility of the structures to sterically hinder potential clashes and/or to optimize the energy values. The structures obtained were immersed in a cubic box filled with TIP3P water molecules, and the system was neutralized with the addition of counter ions. Simulations were run by applying periodic boundary conditions. Energy minimization was performed with 5000 steps using steepest descent as the algorithm, which converged to a minimum energy with forces less than 10 kJ/mol/nm. A short 25 ns classic molecular dynamics (cMD) simulation was performed to relax the system. All the cMD simulations were performed by integrating each time step of 2 fs; a Nose–Hoover thermostat maintained the temperature at 300 K, and a Parrinello–Rahman barostat maintained the system’s pressure at 1 atm, with a low dumping of 1 ps^−1^; the LINCS algorithm constrained the bond lengths that involved hydrogen atoms.

### 4.5. Virtual Screening

To reinforce the reliability of our simulations, we applied a docking simulation based on in vitro evidence; thus, we only selected the targets in which their experimental 3D structures were combined with an active compound (if more 3D structures of the same target were combined with different ligands in different binding regions, like allosteric pockets, a box able to enclose such binding regions was created). Thus, a box with dimensions of 25 Å × 25 Å × 25 Å was created, and we set the grid box to the ligand’s center of mass in the experimental 3D structure of the target using AutoDock Tools v.4.2 [35,36]. To provide a more consistent result for our docking simulation, we changed the default exhaustiveness from 8 to 32 and only selected binding poses with a RMSD that was 2 Å lower than that of the best docked pose. All other parameters were used as default.

Then, a virtual screening using the compounds extracted via chestnut burr biomass was carried out on targets using AutoDock/VinaXB v.1.1.2. MGLTOOLS v.1.5.7 scripts and OpenBabel v.3.1.0 [37,38] were used to, respectively, convert protein and ligand files and to add gasteiger partial charges. Three-dimensional structures of gallic acid, quinic acid, protocatechuic acid, brevifolin carboxylic acid, and ellagic acid were retrieved from the PubChem database [39] (with compound CID: 370, 6508, 72, 9838995, and 5281855, respectively). The interaction network was explored with the P.L.I.P. v. 2.3.0 Tool. The alignment of sequences, which is useful for the identification of the key residues of targets, was performed with ClustalW v.2.1 [40].

### 4.6. Steered Molecular Dynamics (SMD) Simulations

The unbinding pathway of complexes obtained with the docking simulation was simulated by applying a SMD run of 800 ps of a constant pulling force of 150 kJ/mol/nm. The target backbone was not allowed to move, and the compound was pulled with an external force in the NPT ensemble at 1 atm and 310 K with 2 fs time steps, with a constant force in the x, y, and z directions. All MD analyses were performed with the GROMACS 2019.3 package and displayed with GRACE v.5.1.25. PyMOL 2.5 was used as the graphical interface and generated the pictures. All simulations were run on LINUX Mint 21.1 Cluster with 660 CPUs on 21 different nodes, 190T of RAM, 30T hard disk partition size, and 6 NVIDIA TESLA GPU with CUDA support (available at the Department of Biotechnology, Chemistry, and Pharmacy of Siena University).

## 5. Conclusions

Several natural sources, like plants or microorganisms, represent a valid alternative to explore potential bioproducts that are similar to medicine. In fact, they provide a huge opportunity for us to identify and isolate new compounds that possess biological activities.

Their beneficial proprieties are known for thousands/hundreds of years and have been traditionally used in traditional Chinese medicine. They can be obtained from natural sources and are able to influence wide signaling pathways, like cell proliferation, survival, death, angiogenesis, etc. They also possess antimicrobial, antioxidant, and anti-inflammatory properties.

Here, we applied a novel drug and target discovery approach developed within the circular bioeconomy concept, and we extracted and identified antioxidant bioproducts from the chestnut burr biomass (agro-forest waste). Then, their potential biological targets were identified.

Overall, our findings provide strong evidence that ellagic acid is a promising compound for the inhibition of the kinase family.

In recent years, the kinase inhibitors are receiving broader interest in the midst of the search for new drugs, representing promising candidates for effective drugs against severe diseases. Many kinases are targeted by natural compounds, which are extracted from different natural sources [41,42].

Ellagic acid is a phenolic compound widely present in several red fruits and berries. Remarkable biological properties are known about ellagic acid, like its antioxidant [43], anti-inflammatory [44], antimicrobial [45], antidiabetic [46], antiviral [47], antidegenerative [48], and anticancer [49] potential. Also, several studies have extensively reported the use of ellagic acid in topical and systemic applications. Ellagic acid has further been proposed as a compound with effective proprieties for skin tumors, dermatitis, cutaneous leishmaniasis, as well as photoprotective [15] and antiaging agents, providing it with suitable proprieties for the prevention and treatment of skin disorders.

Despite the great advantages of ellagic acid, the application of this compound is limited due to its chemical-physical proprieties, like its low permeability and low solubility in aqueous solvents. However, in the recent decade, several approaches have successfully been proposed to overcome these drawbacks.

Thus, ellagic acid can be further exploited as a supporting lead for the design of potent and selective molecules against the kinase protein family for the management of oxidative stress conditions as well as other disorders.

Our work provided knowledge about the chemical composition of chestnut burr biomass, suggesting an innovative use of natural waste as a source of potential bioactive compounds targeting biological systems involved in the onset pathologies.

To conclude, our approach was developed within the circular bioeconomy concept integrated in drug and target discovery, in which a green and sustainable scientific methodology was utilized to identify and propose compound-like drug(s) (and their potential target(s)) extracted not only from natural sources but also from agro-forest waste (a pollution source), remarkably reducing time, costs, and chemical waste, and thus addressing the tragical impact of pollutants on our planet’s environment.

## Figures and Tables

**Figure 1 ijms-25-02517-f001:**
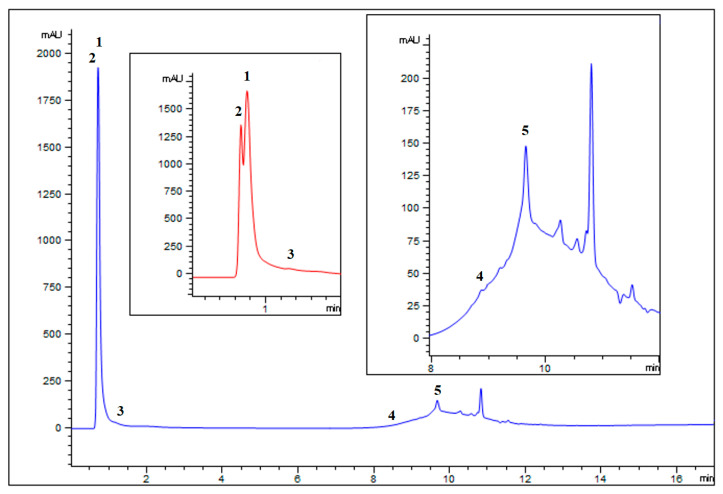
Chromatographic profile of chestnut burrs extract obtained at 254 nm (blue) and 210 nm (red). Analysis of the chestnut burr extract under the conditions described in the experimental section (in the insets details of the chromatograms). (1) Gallic acid, (2) quinic acid, (3) protocatechuic acid, (4) brevifolin carboxylic acid, and (5) ellagic acid. Their identification was ascertained by comparing retention times with standard compounds.

**Figure 2 ijms-25-02517-f002:**
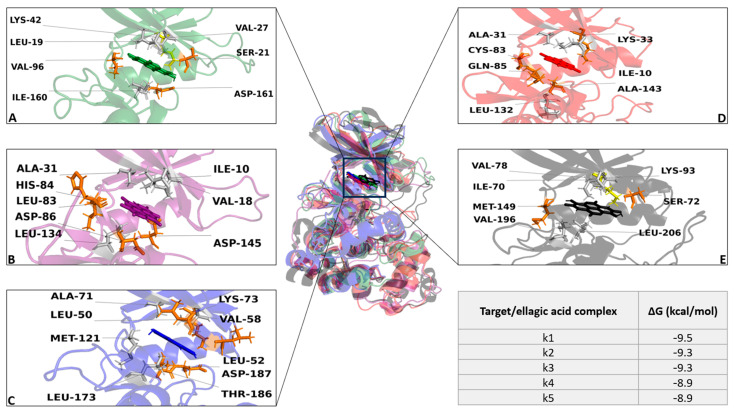
Overview of target/ellagic acid complexes. In the image, 3D structures of k1 (green), k2 (purple), k3 (blue), k4 (red), and k5 (black) are reported in the same complex with ellagic acid (in sticks). The enlarged pictures (**A**–**E**) show the interaction network formed among target binding residues and ellagic acid. The binding residues involved in hydrophobic interactions, hydrogen bonds, and salt bridges are represented as gray, orange, and yellow sticks, respectively. The table displayed on the bottom right side shows the binding free energy (kcal/mol) of the ellagic acid in combination with the targets.

**Figure 3 ijms-25-02517-f003:**
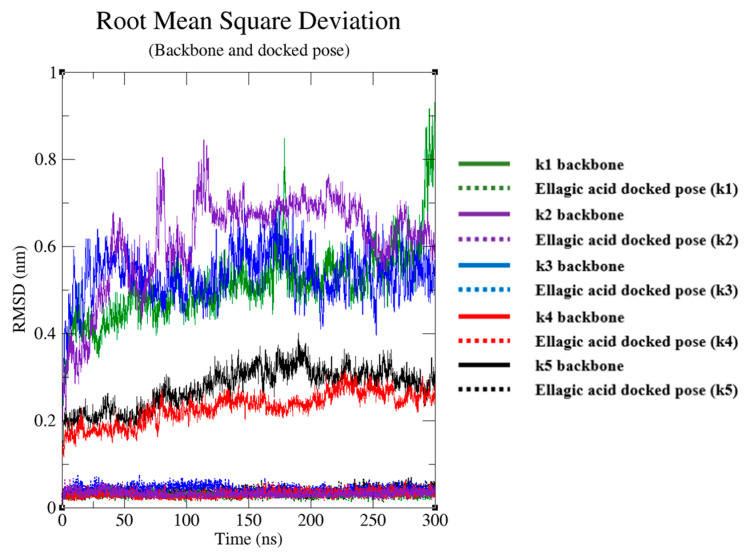
Root mean square deviation (RMSD) plots. The RMSD profile of ellagic acid’s docked pose (dotted line) and protein backbone (continue line), relative to the initial frame against simulation time.

**Figure 4 ijms-25-02517-f004:**
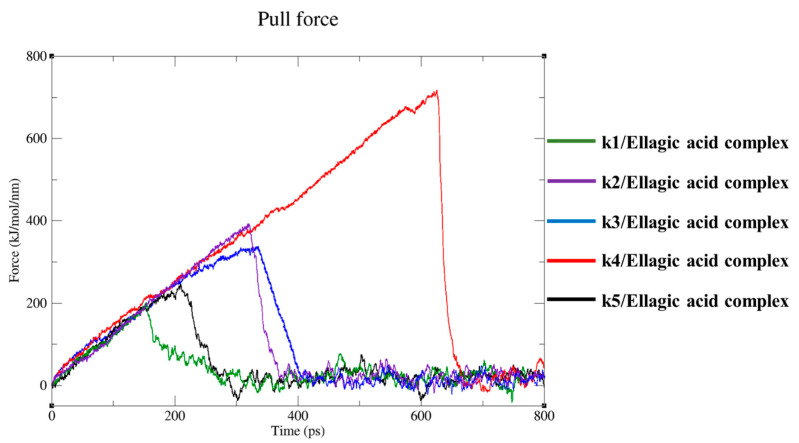
Steered molecular dynamics simulations. Force profiles of ellagic acid pulled out of the target binding pocket along the unbinding pathway.

## Data Availability

Data is contained within the article and Appendix A.

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
