# Peer review of "A Drug Discovery Approach to a Reveal Novel Antioxidant Natural Source: The Case of Chestnut Burr Biomass"

_ijms, 2024, doi:10.3390/ijms25052517_

Round 1
Reviewer 1 Report
Comments and Suggestions for Authors
I think the manuscript cannot be accepted in the current format as there are some major issues that need to be addressed to support the study.
1. In the LC-MS experiment, the authors identified five compounds. However, the authors didn't provide any mass spec results to prove the identity of the compounds. In addition, the chromatographic profile looks a bit confusing in Figure 1. I only see a major peak in 1/2 and some lower but wide peaks with for 5. There is no peaks for 3 and 4. There are also some spikes in 5, do they mean there are more than one molecules at that position?
2. The authors did in sillico docking to identify potential target of those compounds. However, it is unclear to me how the authors identified the binding pockets at the first place. For those kinase, are there studies showing similar compounds binding to the same pockets? Could there be other allosteric pockets?
3. The authors also did MD simulations to predict the docking pose and binding energy. However, the authors didn't provide any control simulation such as docking the compounds to the wrong pockets or docking a non-specific molecule and perform MD simulation. Therefore, without further control, the MD results cannot serve as a strong proof of the binding.
In general, I think this study lack convincing data and control experiments to show that these molecules could be useful for drug discovery.
Comments on the Quality of English LanguageThe English needs to be further improved.
Reviewer 2 Report
Comments and Suggestions for Authors
This work by Trezza et al. explores a novel approach within the circular bioeconomy concept to extract and identify antioxidant bioproducts from chestnut burr biomass, an agro-forest waste, targeting the drug and target discovery field. It highlights the discovery of ellagic acid, a phenolic compound found in red fruits and berries, known for its extensive biological properties including antioxidant, anti-inflammatory, and anticancer activities, among others. Despite its beneficial qualities, ellagic acid's application has been limited by its low solubility and permeability. However, recent advancements have overcome these challenges, positioning ellagic acid as a promising scaffold for designing potent molecules against the kinase protein family, addressing oxidative stress and other disorders. This research not only uncovers the potential of using natural waste as a source of bio-active compounds but also promotes a sustainable, cost-effective, and environmentally friendly methodology in the drug discovery process. This finding is significant as it opens new avenues for drug development, leveraging waste materials for therapeutic purposes, thus contributing positively to both healthcare and environmental sustainability.
A few points the authors could consider in revising their manuscript are listed below:
The detailed comments are as follows:
1) How was the molecular modeling performed to optimize the 3D structures of the targets? Were any specific software or algorithms used, and how were they validated to ensure the accuracy of the structures obtained?
2) The SMD simulations report a varied unbinding path for the compound across different kinase targets (k1, k2, k3, k4, k5) with specific mention of the initial slow detachment phase and a second phase where the compound moves away from the target. Can you explain the biological significance of these variations in unbinding times and forces among different targets? How does this variability affect the compound's potential efficacy against these targets?
3) How did these simulations confirm the reliability of the docked pose and the interaction energies with the target?
4) Can you elaborate on the virtual screening process used to identify the bioactive compounds with antioxidant properties and their biological targets? What parameters were set for the docking simulations, and how was the binding free energy score determined to be significant?
5) Regarding the LC-MS analyses for experimental characterization, could you specify the conditions under which these analyses were conducted, including the preparation of the chestnut burr biomass extracts, the LC-MS settings, and how ellagic acid was quantified as the most abundant component?
6) Figure S3, Caption is not explained properly.
Finally, once the above comments are fully addressed, the manuscript could be accepted for publication in this journal.
Round 2
Reviewer 1 Report
Comments and Suggestions for Authors
The author did addressed my comments regarding the in sillico docking and MD simulation. However, without providing further proofs of all the compound identities such as Mass Spec Spectroscopy and/or NMR Spectroscopy, I am not confident on the identities of the compounds (I understand that the author compared the retention time to standard compounds). The small molecule identities are the basis of this current study, therefore more rigorous and convincing evidence need to be provided.
Comments on the Quality of English LanguageN/A
Round 3
Reviewer 1 Report
Comments and Suggestions for Authors
The author claimed that the compound identities are confirmed based on the mass signals (pseudomolecular ions). However, I didn't see any mass spectra provided in the main manuscript or in the supplementary. Unless the authors can provide the raw mass spectra highlighting the mass signals for each individual compounds and for all the target peaks on the chromatographic profiles, it is really difficult for me to accept the paper in the current form. Again making sure the identities of these compounds is really important as they are the fundations of this studies. The authors should provide these data if they claimed they have done so.
Comments on the Quality of English LanguageN/A
Round 4
Reviewer 1 Report
Comments and Suggestions for Authors
The author provides evidence to address my comments. The paper can be accepted in the current form after checking typo etc.
Comments on the Quality of English LanguageN/A